# In Vitro Antibacterial Mechanism of High-Voltage Electrostatic Field against *Acinetobacter johnsonii*

**DOI:** 10.3390/foods11070955

**Published:** 2022-03-25

**Authors:** Han Huang, Tianqi Gao, Xiaoqing Qian, Wenjing Wu, Xiuzhi Fan, Liu Shi, Guangquan Xiong, Anzi Ding, Xin Li, Yu Qiao, Li Liao, Lan Wang

**Affiliations:** 1Key Laboratory of Agricultural Products Cold Chain Logistics, Ministry of Agriculture and Rural Affairs, Institute of Agro-Products Processing and Nuclear agricultural Technology, Hubei Academy of Agricultural Sciences, Wuhan 430064, China; 201775166@yangtzeu.edu.cn (H.H.); gaotianqi7739@163.com (T.G.); qianxq1999@163.com (X.Q.); 272081603@163.com (W.W.); xzhfan@163.com (X.F.); shiliu_hzau@163.com (L.S.); xiongguangquan@163.com (G.X.); sw03@163.com (A.D.); leexin117@163.com (X.L.); qiaoyu412@sina.com (Y.Q.); liaoliss@126.com (L.L.); 2State Key Laboratory of Food Science & Technology, Jiangnan University, Wuxi 214122, China; 3School of Food Science and Technology, Jiangnan University, Wuxi 214122, China; 4Department of Biomedicine and Biopharmacology, Bioengineering and Food College, Hubei University of Technology, Wuhan 430068, China

**Keywords:** *Acinetobacter johnsonii*, high-voltage static electric field, biochemical properties, stress-related genes

## Abstract

This study aimed to investigate the antibacterial properties and mechanisms of a high-voltage static electric field (HVEF) in *Acinetobacter johnsonii,* which were assessed from the perspective of biochemical properties and stress-related genes. The time/voltage-kill assays and growth curves showed that an HVEF decreased the number of bacteria and OD600 values. In addition, HVEF treatment caused the leakage of cell contents (nucleic acids and proteins), increased the electrical conductivity and amounts of reactive oxygen substances (ROS) (16.88 fold), and decreased the activity of Na+ K+-ATPase in *A. johnsonii*. Moreover, the changes in the expression levels of genes involved in oxidative stress and DNA damage in the treated *A. johnsonii* cells suggested that HVEF treatment could induce oxidative stress and DNA sub-damage. This study will provide useful information for the development and application of an HVEF in food safety.

## 1. Introduction

Antimicrobial technologies are essential for extending food storage time due to their bacteriostatic effect in reducing the contamination risk of spoiling and pathogenic bacteria [1,2,3]. After sterilization, food is relatively safer. In food, physical sterilization is regarded as safer than chemical sterilization; thus, the development of safe physical antimicrobial technology has gained considerable interest [4,5,6].

Electrostatic field treatment has a high efficiency and low energy consumption without chemical residues [7,8,9]. There are three available discharge modes, including high-voltage direct current (DC), high-voltage alternate current, and high-voltage pulsed current. The strong antibacterial effect of a pulsed electric field (PEF) has been extensively studied. Inactivation is achieved through different plasma sources, such as atmospheric-pressure air plasma (DC) and cold atmospheric-pressure plasma, which effectively decrease the total plate counts of tilapia (*Orechromis niloticus*) fillets [4]. Xu and Cheng [10] reported that the bacterial viable count of *Staphylococcus aureus* (*S. aureus*) and *Escherichia coli* (*E. coli*) decreased from 7 log10 CFU/mL to 1.65 log10 CFU/mL and 0.44 log10 CFU/mL when exposed to DC air plasma for 20 min, respectively. Hertwig and Leslie [11] explored the inactivation efficiency of *Salmonella Enteritidis* PT 30 in different gas plasma as follows: air > O_2_ > N_2_. Ahmad Shiekh and Benjakul [5] found that the mesophile spoilage bacteria growth and sensory quality deterioration of cold-stored Pacific white shrimp (*Litopenaeus vannamei*) could be reduced by combining PEF (15 kV/cm, 600 pulses, 483 kJ/kg) with chamuang leaf extract (*Garcinia cowa*
*Roxb*., 0.5–1%).The synchronization of PEF (3 times under 8T, 60 pulses) and *Litseacubeba* essential oil could significantly damage the membrane and intracellular enzymes of *Escherichia coli* O157:H7 [12]. An HVEF could also effectively inactivate *Staphylococcus aureus* in four food systems (salmon, griskin, cheese and sausage); the bacterial inactivation mechanism was similar with PEF [7]. In addition, an HVEF had an even voltage distribution and was much more stable than DC, PEF, and a high-voltage pulse [13]. There was little electric current in the HVEF process, and the energy consumption was lower than other treatments [14,15].

*Acinetobacter**,* as obligate aerobes, easily survive in a humid environment, especially growing in the water and soil surrounding fish cultivation [16,17]. They are considered to be opportunistic pathogenic bacteria, representing the G^-^ non-fermented microorganisms encountered in clinical medicine, and are often discovered in nosocomial infections, especially in intensive care units [18,19]. Moreover, they are usually discovered in fresh, spoiled, and processed foods [20,21]. In our previous study, it was proved that HVEF treatment could inhibit microbial growth, change the bacteria phase distribution of fresh and spoiled channel catfish, and reduce the relative abundance of *Acinetobacter* [22]. *Acinetobacter radioresistens* was isolated and identified from fresh channel catfish [23]. HVEF treatment resulted in the membrane protein degradation of *A. radioresistens* and activated anti-oxidation responses. Subsequently, *Acinetobacter johnsonii* was isolated from the fish too. The difference between the effect of HVEF on *A. johnsonii* and *A. radioresistens* is unknown. In this manuscript, in order to deepen the research on the preservation mechanism of an HVEF on channel catfish, we focused on the inactivation mechanism of an HVEF in *Acinetobacter johnsonii*. Therefore, flow cytometry (FCM), combined with a fluorescent transmission electron microscopy (TEM), scanning electron microscopy (SEM), and real-time quantitative PCR techniques, was used to elucidate the effect of an HVEF against *Acinetobacter johnsonii*. This study could provide more information on the inactivation mechanism of an HVEF and contribute new perspectives to the application of electric field in food sterilization.

## 2. Materials and Methods

### 2.1. Materials

*A. johnsonii* (MW362499), isolated from fresh channel catfish (Wuhan Baishazhou fresh market, China), was stored in the institute for Farm Products Processing and Nuclear-Agricultural Technology, Hubei Academy of Agricultural Science. The chemical agents were analytical grade from Sinopharm group.

### 2.2. HVEF Treatment

The HVEF equipment was collected as previously described [22]. The bacterial samples were put on a baseplate, 7 cm away from the top plate. The optimal condition of HVEF sterilization treatment was set as 30 kV/15 min. *A. johnsonii,* treated with HVEF at 30 kV/15 min, was set as T group, while the control group was set as C group.

### 2.3. Bacterial Growth Curve

The growth curve was used to investigate the bactericidal effects of HVEF treatment against *A. johnsonii* according to a procedure described by Bouyahya and Abrini [24]. The voltage-dependent and time-dependent bactericidal activities of HVEF were determined in advance. Then, *A. johnsonii* was cultivated at 37 °C for 0, 1, 2, 3 to 21 h, respectively. At the designated time intervals, the optical density (OD) of supernatants was measured by a L5S UV/VIS spectrophotometer (Shanghai INESA Analysis Instrument Co., Shanghai, China). All measurements were carried out in triplicate. Afterwards, the growth curve of *A. johnsonii* was developed with time as the horizontal axis (0 to 21 h) and the OD 600 nm of the supernatant as the vertical axis.

### 2.4. Cell Leakage Determination through UV Absorption

Bacterial membrane integrity was determined with a L5S UV/VIS spectrophotometer, as previously described [3]. The samples of two groups were centrifuged at 6000× *g* for 10 min. The OD_260_ and OD_280_ of the supernatant were measured as the leakages of nucleic acid and protein, respectively.

### 2.5. Cell Membrane Permeability Determination

The conductivity on the cell membrane surface was measured by a previously reported method with some modifications [25]. Cell membrane permeability was characterized by a conductivity change in the culture medium. Bacteria liquids from two groups were diluted, and their conductivity was measured with a FE28 conductivity meter (Mettler-Toledo, Leicester, UK).

### 2.6. Activity of Na^+^ K^+^-ATPase

The activity of Na^+^ K^+^-ATPase was detected to reveal the damage levels of cell membranes and was measured following earlier publications [2]. The ultra-micro activity of Na^+^ K^+^-ATPase of each sample was tested with a rapid test kit (Jiancheng Bio-technology Co., Ltd., Nanjing, China).

### 2.7. Intracellular ROS Concentration

Intracellular ROS concentration was quantitatively estimated following the method of Tang and Liu [26]. ROS generation was determined using DCFH-DA, a ROS indicator producing fluorescent DCF in the presence of intracellular oxygen. *A. johnsonii* cell collection was performed as previously described [23]. *A. johnsonii* cells from two groups were incubated in 1 mL of 10 mM 20′,7′-dichlorodihydrofluorescein diacetate (DCFH-DA, Beyotime, China) at 37 °C for 20 min. The extracellular DCFH-DA was removed by repeatedly washing cells. The fluorescence of the intracellular DCF was determined using CytoFLEX flow cytometry (Beckman Coulter, Inc., Fullerton, CA, USA). ROS generation was evaluated by comparing fluorescence of treated and untreated cells.

### 2.8. SEM, TEM Observation and Negative Staining of Bacterium

SEM: *A. johnsonii* was inoculated into LB liquid medium and incubated in a shaker at 37 °C for 24 h to log phase. Then, the cells were centrifuged (6000× *g*, 10 min) and washed with 0.1 M PBS 3 times (6000× *g*, 10 min) to avoid medium residue. Subsequently, 2.5% (*v/v*) glutaraldehyde fixative was added to the cells, fixed in a refrigerator at 4 °C for 3–4 h, and the cells were washed 3 times in 0.1 M PBS (6000× *g*, 10 min). Then, different concentrations of ethanol solution (50%, 70%, 80%, 90%, 100%, *v/v*) were added into the cell pellet to dehydrate it for 15 min, and cell suspension was centrifuged (6000× *g*, 10 min). Subsequently, the dehydrated cell sample was immersed in isoamyl acetate for 30 min at room temperature, and the cells were subjected to CO_2_ critical drying. Finally, the dried powdered pure cells were sprayed with gold, and the morphology of the cells was observed through the SIGMA 300 [7].

TEM: TEM observation of *A. johnsonii* was obtained from Lin and Wang [12] with slight modifications. The cells were centrifuged (6000× *g*, 10 min), washed in 0.1 M PBS (pH 7.0) 3 times and resuspended in 0.1 M PBS. Subsequently, the cells were fixed with 2.5% (*v/v*) glutaraldehyde solution at 4 °C overnight. Then, the cells were collected by centrifugation (6000× *g*, 10 min, 4 °C) and washed 3 times with 0.1 M PBS (pH 7.0). Finally, all samples were observed in TEM of Model JEM-1230.

Negative staining of bacterium: The bacterium pretreatment was performed as described previously in TEM treatment. The bacterium suspension was stained by 1% to 2% solution of phosphotungstic acid for 30 s. Then, all samples were observed in TEM of Model JEM-1230.

### 2.9. Gene Expression Analysis

*A. johnsonii* cell collection and RNA extraction were performed as previously described [23]. Appropriate amount of cell sample was placed into 1.5 mL Eppendorf tube, 1 mL Trizol reagent was added into tube, and the two components were mixed. Chloroform was added according to the ratio of Trizol: chloroform 5:1. The three components were violently shaken for 15 s, left to stand at room temperature for 3 min, and then centrifuged (12,000× *g*, 4 °C, 15 min). The supernatant was absorbed into another 1.5 mL Eppendorf tube, and isopropyl alcohol was added in equal volume. After mixing, the tube was left to stand at room temperature for 20 min, centrifuged (12,000× *g*, 4 °C, 10 min), and then the supernatant was removed. At least 1 mL of 75% precooled ethanol was added into tube at 4 °C, and then the sediment was washed in ethanol and precipitated again. After centrifuging at 4 °C at 10,000× *g* for 5 min, the supernatant was discarded. After centrifuging again at 4 °C for 5 min at 10,000× *g*, residual liquid was removed, and the sediment was dried at room temperature. Then, 20 µL RNASE-free water was added until it was completely dissolved, and 1 µL solute was taken for electrophoresis detection. Retro-transcription amplification was performed using the Goldenstar RT6 cDNA Synthesis Kit. The cDNA product, obtained by reverse transcription, was added to 10 uL ddH_2_O and diluted to 30 ul for qPCR template, which was amplified with the Optimus 2 × T5 Fast qPCR Mix (SYBR Green I). Specific primers sequences for qRT-PCR were as follows: *AhpC* (5′F: GGCATCCAGTCTAACTTTGACGT; R: GGCCTTGACCGATATGGTTATT), *KatE* (5′F: GGACTCCAAAGCAAGGGGTA; R: TTCCGGGTAATTGCCACG), *Nfo* (5′F: GCACATTGGGAACAAGCCTT; R: GAATCGAAGGAATTTCATCGCT), *SodA* (5′F: CCGCACATCAGCAAAGAAACT; R: GCAGAAGCTGTGATGATTTCTTCTA), *RecG* (5′F: TTGTGGCGAATGCTGTGAGT; R: GGCAGGGTAGAGGTGTTTTTGT), *RadA* (5′F: ACGGGTTCCGTGGTACTGAT; R: GTTTATCTGTGGGTAGATCAAGACG), *RecN* (5′F: CTGAATTGAAAGAAATCGGGC; R: ATGTGCTATAGGCTTCACGCAC), *Dps* (5′F: GCAAAGTATCTTAGTGACAGTACTCATTG; R: CTGTTGAGTGAGTGCATTTGAGAC). The relative mRNA expression, calculated from the fluorescence signal, was displayed by the 2^−∆∆Ct^ algorithm, and quantified by the internal reference gene [27].

### 2.10. Statistical Analysis

Statistical analysis was performed using SPSS 20 software (SPSS Inc., Chicago, IL, USA). The paired sample *t*-test was used to determine the significant difference between treatment group and control group; *p* < 0.05 was considered as statistically significant. Graphs were drawn with GraphPad Prism 5 software (GraphPad Software Inc., San Diego, CA, USA). All the experiments were conducted in triplicate.

## 3. Results and Discussion

### 3.1. Effect of an HVEF on A. Johnsonii Growth

In order to verify the antibacterial effect of the HVEF, the time-kill assays, voltage-kill assays, and growth curve of *A. johnsonii* in the control group and the treatment group were plotted. Compared with other voltages and electric field treatment times, the number of bacteria was the smallest in the treatment group (T) (Figure 1a,b). It can be seen from the results of the growth curve (Figure 1c) that *A. johnsonii* in the control group (C) showed a good growth activity, and the OD_600_ value was significantly higher than that of *A. johnsonii* in the treatment group (T), indicating that the HVEF had an obvious inhibitory effect on *A. johnsonii*. This result indicated that the inhibition of the HVEF in the growth of *A. johnsonii* occurred through the whole growth period. Our results were consistent with previous study findings of an HVEF used against *L. monocytogenes* [28,29,30]. In addition, compared with previous research results [23], it was found that the bacterial growth ability of *A. johnsonii* was higher than that of *A. radioresistens*. Compared with the voltage–kill assays of two species of *Acinetobacter*, the decrease in the number of colonies of *A. johnsonii* (2.06 Log CFU/mL) with a 30 kV/15 min electric field treatment was higher than that of *A. radioresisten* (1.77 Log CFU/mL), indicating that HVEF treatment had a stronger antibacterial ability against *A. johnsonii*.

### 3.2. Cell Leakage Determination by UV Absorption

The OD_260_ and OD_280_ values represent the leakage of nucleic acid and protein, respectively [31]. Figure 2a shows the changes in OD_260_ and OD_280_ values. After HVEF treatment, the nucleic acid and protein contents of *A. johnsonii* in the supernatant significantly increased by 1.18 and 1.11 times (*p* < 0.05), respectively. The OD_260_ and OD_280_ values of *A. johnsonii* in the treatment group (T) were higher than those in the control group (C), indicating that the cell membrane integrity of *A. johnsonii* was destroyed after HVEF treatment. Pan and Cheng [8] reported that the OD_260_ value increased through the change in the transmembrane transport process of Listeria monocytogenes with the extension of the electric field treatment time. Compared with the results of our previous studies, HVEF treatment increased the OD_260_ value of *A. johnsonii* by 1.17 times and increased the OD_260_ value of *A. radioresistens* by 1.11 times [23]. This indicated that an HVEF might damage the cell membrane and increase the permeability of the cell membrane, resulting in a greater leakage of nucleic acid and protein. The amount of nucleic acid leakage of *A. johnsonii* was larger than that of *A. radioresistens*.

### 3.3. Electrical Conductivity (ETC) and Na^+^ K^+^-ATPase Activity of Cell Membrane

Electrical conductivity experiments with *A. johnsonii* showed an increase in electrical conductivity (ETC) after HVEF treatment (Figure 2b (left)). The ETC value of *A. johnsonii* in T group increased to 14.91 μs/cm, while the ETC value of the C group was 13.92 μs/cm, with a significant difference (*p* < 0.05) between the two groups. Our results were consistent with one previous study reporting the change in ETC in *E. coli* treated with cold plasma [32]. These results suggested that the membrane permeability of the bacteria was changed by HVEF treatment, resulting in the leakage of metal ions from the cell membrane to the medium. 

ATPase, acting as an ion exchanger, co-transporter, and pump, inputs many substances needed for cell metabolism and outputs metabolic wastes, which are important for maintaining the resting potential of the cell membrane [33]. Establishing electrochemical gradients across the plasma membrane is critical for cells to perform physiological functions, such as signal transmission and nutrient uptake [12]. Figure 2b (right) shows that the Na^+^ K^+^-ATPase activity of T group (78.63 U/mg pro) was found to be much lower than that of C group (174.12 U/mg pro), suggesting the HVEF treatment decreased the activity of Na^+^ K^+^-ATPase. The above results indicated that HVEF treatment increased the permeability of the cell membrane by significantly damaging the bacterial cell membrane.

### 3.4. Oxidative Stress Induced by HVEF

Reactive oxygen species (ROS) play critical roles in cell survival, apoptosis, and death [33]. Based on this, it was speculated that HVEF treatment might exhibit antibacterial activity against *A. johnsonii* by inducing the production of ROS. In this study, ROS were investigated by flow cytometry. Intracellular ROS were found to increase by 16.88 fold in the HVEF treatment group when compared to untreated control cells (Figure 3). Our results were in line with previous study findings that electric field treatment enhanced the production of ROS [10]. The high concentration of ROS might hinder the function of some organelles within the cells, thus leading to the destruction of the cell structure [34]. Furthermore, excessive intracellular ROS could induce the release of apoptotic factors from the mitochondria, causing DNA damage [35].

### 3.5. Morphological Changes of A. johnsonii

To determine the effect of HVEF treatment on cell morphology, including the permeability and integrity of cell membranes, the changes in cell morphology were observed by SEM (Figure 4a,b). *A. johnsonii* cells of the C group exhibited a regular short rhabditiform morphology with a smooth and regular surface, and the cells were uniform in size and distribution, with few damaged membranes and complete cellular structures (Figure 4a). However, in the T group, the bacterial cells lost the regular short rhabditiform morphology, with many pits and ruffles occurring on the surface of *A. johnsonii* cells. Moreover, uneven fragments were observed, indicating that the damages were induced on bacterial cell membranes (Figure 4b). These results suggested that HVEF treatment damaged the cell membranes and walls, and cell surface morphological alterations of *A. johnsonii*, including membrane permeability and integrity, might destabilize the bacterial cells. Similar results were reported in one previous study that a loss of *E. coli* cell functionality was induced by electric field treatment [36]. In addition, the collective intracellular ROS from HVEF treatment could induce lipid peroxidation and cause the physical damage and inactivation of *A. johnsonii* cells [23].

The changes in cellular structure caused by HVEF were observed by TEM. TEM images confirmed that *A. johnsonii* subjected to HVEF treatment morphologically changed (Figure 4c,d). Bacterial cells of the C group were uniformly distributed, displaying intact cell walls and membranes with internal constituents evenly arranged and clearly visible (Figure 4c). In the T group, bacterial cells were partially damaged with a severely mutated morphology, disordered internal cell structure, the partial loss of organelles, and observable cell alterations (Figure 4d). In previous studies, although electric field treatment changed the cytoplasmic density and resulted in cell wall degradation, there were still some viable cells [3]. Electric field treatment also resulted in cell lysis, cytosol leakage, and membrane damage with the extended plasma treatment time [29]. From the negative staining results, it can be seen that the morphology of *A. johnsonii* was different from *A. radioresisten*. There were “filaments” outside the cell wall of *A. radioresisten*, which might be biofilms.

### 3.6. Gene Expression Analysis

According to the bacterial gene expression assay results, bacterial stress mechanisms were categorized into six types, namely, oxidative stress, osmotic, DNA damage, membrane damage, and specific stress caused by heavy metals [37]. Considering that the HVEF treatment displayed an effective antimicrobial action, especially against *A. johnsonii*, we speculated that HVEF treatment might inhibit the expression of the oxidative-stress-response-related genes and DNA damage protection/repair-related genes of *A. johnsonii*. To test this hypothesis, we examined the effects of HVEF treatment on the expressions of four oxidative-stress-response-related genes (Group Ⅰ: *AhpC*, *KatE*, *Nfo*, and *SodA*) and four DNA damage protection/repair-related genes (Group Ⅱ: *RecG*, *RadA*, *RecN*, and *Dps*). The results indicated that oxidative stress was one of the main pathways affected by HVEF treatment.

The expressions of *AhpC*, *KatE*, and *SodA* were up-regulated in T group (Table 1). The *AhpC* expression was found to be up-regulated by 5.98 fold, whereas that of *A. radioresisten* was up-regulated by 1.87 fold [23]. The up-regulated *AhpC* could cause the peroxidase activities of hydrogen peroxide, peroxynitrite, and organic hydroperoxides to exceed the normal level in the bacteria of the T group [38]. The *KatE* expression increased by 1.14 fold, and *KatE* could counteract the increase in the H_2_O_2_ concentrations in the T group. Moreover, the *SodA* expression encoding superoxide dismutase was up-regulated by 65.50 fold, suggesting that HVEF treatment initiated certain responses against oxidative stress at the genetic level. These results indicated that HVEF treatment could increase the concentrations of various oxides, induce oxidative stress, and damage the DNA structure in *A. johnsonii* cells. HVEF treatment showed a great effect on the inactivation and/or death of *A. johnsonii* by up-regulating or down-regulating the related genes.

The mRNA expressions related to DNA structure and function damage also exhibited a difference between the two groups. The *RecG* expression encoding ATP-dependent DNA helicase was up-regulated by 7 folds in T group, compared with C group. Similarly, that of *A. radioresisten* was up-regulated by 1.62 fold [23]. The mRNA expressions of *RadA* encoding one DNA repair protein, *RecN* encoding another DNA repair protein, and *Dps* encoding a DNA-binding ferritin-like family protein were significantly up-regulated by 4.50 fold, 1.88 fold, and 39.30 fold, respectively, in the T group compared with the C group (Table 1), whereas those of *A. radioresisten* were up-regulated by 2.16 fold, 2.92 fold, and 1.23 fold, respectively [23]. These results indirectly confirmed that HVEF treatment activated a regulatory network involving many DNA damage protection/repair-related genes and pathways, and this treatment damaged the DNA structure and function in *A. johnsonii* cells. Compared with the expressions of *AhpC*, *KatE*, and *SodA* of *A. radioresisten*, we found that the expressions of all the genes in both Group Ⅰ and Group Ⅱ were lower than those of *A. johnsonii.* By combining the results of the voltage-kill assays, growth curves, intracellular nucleic acid, and morphological changes of the two Acinetobacter, we concluded that an HVEF had a greater bacteriostatic effect on bacteria without biofilm.

## 4. Conclusions

Our results demonstrated that HVEF treatment appeared to be a potent bacteriostatic technology against *A. johnsonii*. Direct damage to the cell membranes of *A. johnsonii* could explain the antimicrobial mechanism of HVEF treatment. Cell membrane permeability assays showed that HVEF treatment could cause cell membrane rupture and disintegration, resulting in the slow growth, or even death, of bacterial cells. In addition, the transcriptome results showed that HVEF treatment induced significant changes in the expression levels of oxidative-stress-response-related genes and DNA damage protection/repair-related genes. HVEF treatment was found to activate the oxidative stress responses and the DNA damage protection/repair processes in *A. johnsonii.* Hence, the effect of HVEF treatment on *A. johnsonii* might not be acutely lethal, but rather sublethal. A comprehensive consideration of the HVEF-induced damage to *A. johnsonii* and *A. radioresisten* revealed that the damage of *A. johnsonii* was more serious. According to the morphological results, it was hypothesized that the differences in HVEF-induced damage between *A. johnsonii* and *A. radioresisten* might be ascribed to differences in the bacteria biofilms. The impact of an HVEF on the *A. johnsonii* bacteria biofilm needs further study.

In general, the current study provided new insights into correlating electric-mediated alterations in DNA damage and membrane integrity with intracellular ROS levels and bacteriostatic efficiency, suggesting the potential possibility to improve or optimize the sterilization efficiency in real food systems by HVEF treatment. However, more investigations are still needed to acquire a better understanding of the distribution, penetration, and interaction mechanism of HVEF-mediated ROS at a molecular level, from the outside of the cell membrane to the interior, especially in real food systems.

## Figures and Tables

**Figure 1 foods-11-00955-f001:**
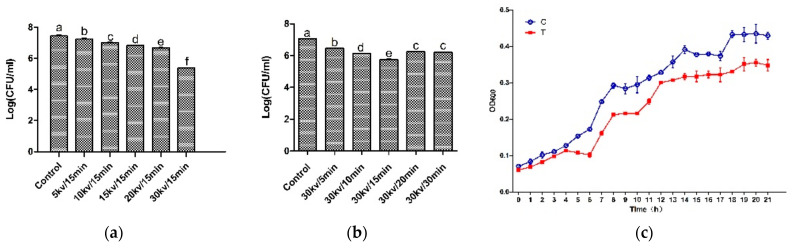
The time–kill assays (**a**), voltage–kill assays (**b**), and the growth curve (**c**) of HVEF against *A. johnsonii*. C: the untreated samples; T: HVEF-treated samples. Different letters (a–f) indicate significant differences between control and HVEF-treated samples. (*p* < 0.05).

**Figure 2 foods-11-00955-f002:**
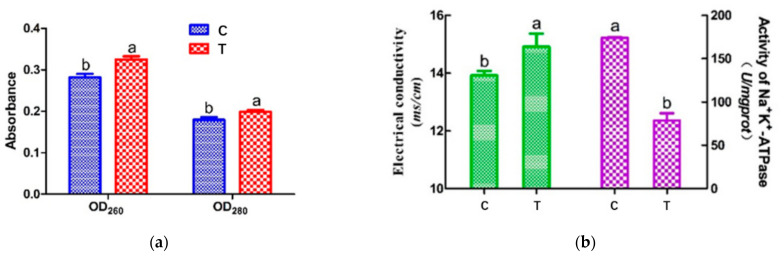
The intracellular nucleic acid (260 nm) and protein (280 nm) leakage (**a**) and the electrical conductivity and activity of Na^+^ K^+^-ATPase (**b**) in *A. johnsonii* after HVEF treatment. Different letters (a, b) indicate significant differences between control and HVEF-treated samples. (*p* < 0.05).

**Figure 3 foods-11-00955-f003:**
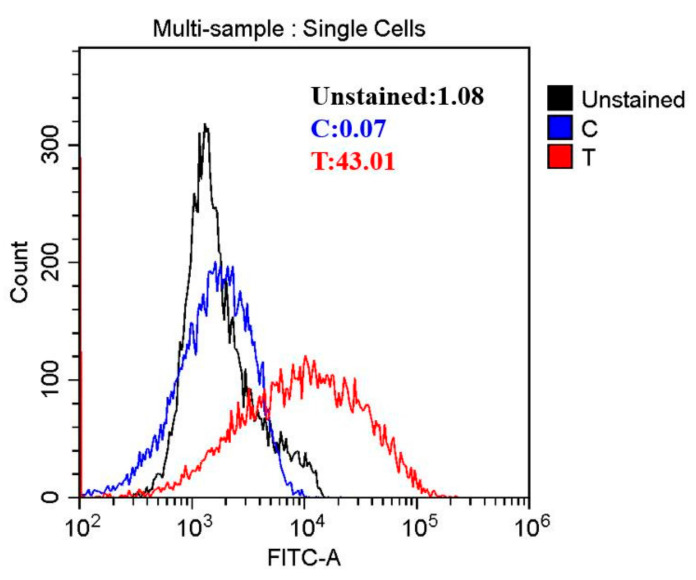
The reactive oxygen species (ROS) of *A. johnsonii* after HVEF treatment.

**Figure 4 foods-11-00955-f004:**
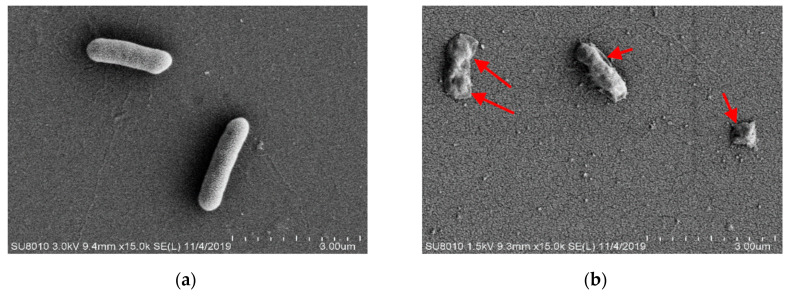
SEM pictures of *A. johnsonii* in (**a**) C group and (**b**) T group. TEM pictures of *A. johnsonii* in (**c**) C group and (**d**) T group. Negative staining pictures of *A. johnsonii* in (**e**) C group and (**f**) T group.

**Table 1 foods-11-00955-t001:** The effect of HVEF treatment on relative expression of the target gene of *A. johnsonii*.

Functional Class	Description	C	T
Group Ⅰ Oxidative Stress Response	*AhpC*	33.22 ± 19.22 ^b^	198.62 ± 141.87 ^a^
	*KatE*	72.40 ± 18.57 ^a^	82.68 ± 13.56 ^a^
	*Nfo*	/	/
	*SodA*	20.81 ± 4.90 ^b^	1363.17 ± 167.52 ^a^
Group Ⅱ DNA Damage Protection/Repair	*RecG*	100.28 ± 29.61 ^b^	702.27 ± 151.29 ^a^
	*RadA*	16.25 ± 1.79 ^b^	73.10 ± 7.64 ^a^
	*RecN*	143.55 ± 33.46 ^b^	269.88 ± 127.55 ^a^
	*Dps*	12.59 ± 4.18 ^b^	494.88 ± 159.48 ^a^

C: the untreated samples; T: HVEF-treated samples. Different letters (a, b) indicate significant differences between control and HVEF-treated samples. (*p* < 0.05).

## Data Availability

Research data are not shared.

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
