# Peer review of "In Vitro Antibacterial Mechanism of High-Voltage Electrostatic Field against *Acinetobacter johnsonii"

_foods, 2022, doi:10.3390/foods11070955_

Round 1

Reviewer 1 Report

The authors did extensive efforts in evaluating the effectiveness of HVEF in bacteria Acinetobacter johnsonii. However the authors still miss a very important element: the involvement of foods, especially when the authors intend to publish in the journal “FOODS”. This is because eventually this is an application driven study intending to apply HVEF technology in food treatment, a treatment of pure bacteria culture can be tremendously different with treatment of real foods. One single bacteria strain, although isolated from foods, is also not representative of the microflora of the foods.

Author Response

The manuscript has been not formatted according to journal template, structure is different and references not in the required style. Cited references not in accordance to journal requirements.

Response: Thanks for your suggestion, the format of references throughout the text has been checked and revised.

The abstract needs to be re-written, not well structured.

Response: Thanks for your suggestion, the abstract has been re-written.

The article is well structured and documented, experimental procedure is justified.

The results are presented in a concise manner and sustained by proper figures and tables. Furthermore, the statistical analysis indicates that the results obtained are statistically significant.

The discussion section is clear, understandable and in accordance with the results obtained. BUT i would suggest to develop some paragraphs explaining a bit the future perspectives and applications.

Response: Thanks for your suggestion, the new paragraph has been added in the last of the discussion section as suggested (L375-L382).

The conclusion is explained by the results of the experiments performed.

Reviewer 2 Report

The manuscript has been not formatted accodring to journal template, structure is different and references not in the required style. Cited references not in accordance to journal requirements.

The abstract needs to b re-written, not well structured.

The article is well structured and documented, experimental procedure is justified.

The results are presented in a concise manner and sustained by proper figures and tables. Furthermore, the statistical analysis indicates that the results obtained are statistically significant.

The discussion section is clear, understandable and in accordance with the results obtained. BUT i would suggest to develop sme paragraphs explaining a bit the future perspectives and applications.

The conclusion is explained by the results of the experiments performed.

Author Response

The authors did extensive efforts in evaluating the effectiveness of HVEF in bacteria Acinetobacter johnsonii. However, the authors still miss a very important element: the involvement of foods, especially when the authors intend to publish in the journal “FOODS”. This is because eventually this is an application driven study intending to apply HVEF technology in food treatment, a treatment of pure bacteria culture can be tremendously different with treatment of real foods. One single bacteria strain, although isolated from foods, is also not representative of the microflora of the foods.

Response: Thanks for your suggestion, actually, the manuscript was the continuity study of our last research published in “LWT-Food Science and Technology” (Huang H , Sun W , Xiong G , et al. Effects of HVEF treatment on microbial communities and physicochemical properties of catfish fillets during chilled storage. LWT- Food Science and Technology, 2020, 131:109667), in which we found that HVEF treatment could inhabit microbial growth, change the bacteria phase distribution of fresh and spoiled channel catfish, and reduce the relative abundance of Acinetobacter. In order to deepen the research on the preservation mechanism of HVEF on channel catfish, in this manuscript, we focused on the antibacterial mechanism of HVEF on Acinetobacter johnsonii, which was isolated and identified from channel catfish, so as to provide a theoretical basis for the application of HVEF in food preservation.

Round 2

Reviewer 1 Report

Please then clearly explain in the introduction part that this is a continuity study of your last research published (“LWT-Food Science and Technology” (Huang H , Sun W , Xiong G , et al. Effects of HVEF treatment on microbial communities and physicochemical properties of catfish fillets during chilled storage. LWT- Food Science and Technology, 2020, 131:109667)

Author Response

Response to Reviewer comment

Reviewer #1:

Please then clearly explain in the introduction part that this is a continuity study of your last research published (“LWT-Food Science and Technology” (Huang H , Sun W , Xiong G , et al. Effects of HVEF treatment on microbial communities and physicochemical properties of catfish fillets during chilled storage. LWT- Food Science and Technology, 2020, 131:109667)

Response: Thanks for your suggestion, this part has been added in L81-L83, L88-L90.
